# The Application of a Human Rights Approach toward Crimes of Corruption: Analyzing Anti-Corruption Regulations and Judicial Decisions

**Mahrus Ali [1,*], Andi Muliyono [2] and Syarif Nurhidayat [1]**

[1] Faculty of Law, Universitas Islam Indonesia, Yogyakarta 55584, Indonesia; syarif.nurhidayat@uii.ac.id
[2] Department of Criminal Law, Sekolah Tinggi Ilmu Hukum Manokwari, Manokwari 98312, Indonesia; andi_muliyono@stih.manukwari.ac.id
\* Correspondence: mahrus_ali@uii.ac.id

**Abstract:** This study aimed to examine the connection between the crime of corruption and human rights violations. Indonesia's corruption-eradication regulations have increased the possibility of handling human rights-based corruption cases. This study employed doctrinal legal research that mainly relied on anti-corruption legislation and corruption cases in judicial decisions. The results showed that the law states that corruption infringes on people's economic, social, and cultural rights. We employed a plausible scenario to provide practical explanations of the relationship between the two variables. The types of crimes of corruption have a direct nexus to the violation of human rights. In addition, there was inadequate proof of the connection between corruption and human rights violations in court rulings. Specifically, a few court decisions relate corruption to human rights violations. Judges consider the relationship more thoroughly when making legal considerations and when it is not applied as an aggravated circumstance, resulting in significantly milder prison sentences. The findings imply the necessity of mainstreaming corruption as a human rights violation through comprehensive and massive studies. Furthermore, legal enforcement institutions need to issue guidelines and provide continuous training on handling human rights-based corruption cases to the police, public prosecutors, and judges.

**Keywords:** crimes of corruption; human rights violation; regulation; judicial decision

## 1. Introduction

This study explores the relationship between Indonesian anti-corruption laws and human rights violations. The two variables have a negative nexus, where more corrupt practices reduce human rights protection and fulfillment (Ngugi 2010). Corruption by government officials threatens human rights (Fuhr 2013) and reduces revenue. This lowers the government's ability to fund essential services that determine living standards (Kumar 2008). Additionally, corruption impedes economic growth, heightens poverty, and exacerbates other human rights violations (Spyromitros and Panagiotidis 2022).

It is difficult to identify corruption in Indonesia due to the complex and sophisticated modus operandi (Sadoff 2017; Shams 2001; Delaney 2007; Davis et al. 2015). This crime leads to significant financial losses and infringement on fundamental rights to education, appropriate healthcare, and fair treatment under the law. These conditions prevail in high-profile corruption cases, such as (a) the life insurance case of the Hambalang athletes' guesthouse, (b) the Bank Indonesia liquidity assistance case, (c) the bribery case involving the former Chief Justice of the Constitutional Court, and (d) the electronic identification card case. However, studies on the relationship between corruption and human rights abuse are limited to existing legal regulations. Prabowo (2014) emphasized the importance of comprehending how opportunities, pressure, and justifications for corruption

are presented to potential offenders that often weigh up the perceived rewards and draw-backs. Moreover, Hadiprayitno (2010) analyzed human rights enforcement in the legal and political system. Juwita (2017) examined how international human rights legislation perceived the relationship between corruption and the right to education. Subse-quently, Ersan established that legal understanding and relevant concepts that link human rights and the criminal process help combat corruption (Ersan 2018). According to Farida (2022), confiscating an asset obtained through corruption does not violate human rights.

The present study is aimed at analyzing the nexus between anti-corruption regulations and court decisions in Indonesia and human rights violations. It specifically examines how the types of corruption directed toward government officials and other public servants in both the United Nations Convention against Corruption 2003 (UNCAC) and the Anti-Corruption Act of 1999/2001 infringe human rights. Furthermore, the study examines the judicial responses to the need to apply the human rights approach. The findings of this study contribute to developing an effective corruption strategy on how to incorporate a human rights approach into substantive criminal law and criminal procedures. This can also act as guidance for judges in aggravating the severity of punishment in corruption cases.

The Section 1 highlights the conceptual framework of corruption as a human rights violation. Corruption could be direct, indirect, or remote by triggering other elements, leading to human rights violations. This section also describes several illustrations that establish the nexus between these two variables. The Section 2 discusses the possibility of connecting different types of corruption in the United Nations Convention against Corruption 2003 and the Anti-Corruption Act of 1999/2001 with human rights violations. This section shows that several fundamental rights have been infringed by government officials' bribery, illicit enrichment, trading in influence, and conflict of interest in public procurement. Additionally, the Section 3 analyzes the judicial decisions linking corruption to human rights violations. Several court verdicts show that corrupt practices violating human rights do not aggravate the severity of punishment.

## 2. Material and Method

This study employed doctrinal legal research on corruption cases that relied on anti-corruption legislations and judicial decisions. Primary data were collected from the In-donesian regulatory framework on the types of corruption as promulgated in the United Nations Convention against Corruption 2003 and the Anti-Corruption Act of 1999/2001. At least four and seven types of corruption have been criminalized in the Convention and the Act, respectively. The analysis of these laws focused on the types of corruption crimes and their nexus to the infringement of human rights, and it is followed by descriptive examples. Additionally, court rulings were highlighted to identify the authenticity of the decisions, and to provide sufficient legal considerations to apply the human rights approach to aggravate the severity of punishment. This study also examined corruption in judicial decisions. To prevent the length of explanation and/or analysis, this study collected and re-viewed 100 verdicts in corruption cases obtained randomly from the Directory of Supreme Decisions (putusan3.mahkamahagung.go.id (accessed on 10 February 2023)). However, 38 verdicts dealt with the misappropriation of public finances and extortion perpetrated by judges, civil workers, state apparatuses, advocates, and private parties that infringe human rights violation directly or indirectly. Only nine verdicts showed that corruption directly violates human rights. These data could be used to propose a comprehensive strategy to fight corruption through judicial decisions.

## 3. Corruption as a Human Right Violation: Conceptual Framework

Corruption has no widely accepted definition as its study is multidisciplinary, with different fields developing respective definitions. For instance, political scientists define corruption as the misuse of state power due to the lack of checks and balances. It is charac-terized by sociologists as a lack of socially acceptable norms in nations where divergent

societal ideals cause historical and sociocultural conflicts (Spalding 2014). Furthermore, Reyes defined corruption as the abuse of public or entrusted power for private gain (Reyes 2019), or a breach of duty. Public officials are considered corrupt when they receive money and fail to fulfill their legal obligations (Underkuffler 2013).

Klitgaard stated that Corruption = Monopoly Power + Discretion − Accountability. This formula implies that corruption occurs when a person has a monopoly of power and considerable discretionary authority but is unaccountable to public scrutiny (Terracino 2012). Although this standpoint focuses more on the causes, it attempts to describe the crime. Stuart defined corruption by dividing it into grand and petty categories. Grand corruption is committed by high-level presidents, ministers, and public officials and involves large sums of money. The second definition involves low-level employees with small sums of money, such as low-level civil servants, customs officers, or traffic police.

The two definitions mean that human rights-based corruption is connected to three factors. First, it is only perpetrated by state actors, including the police, presidents, ministers, members of the House of Representatives, the Indonesian National Army, regents, mayors, governors, and civil officials. This is because the state is perceived as the only actor violating human rights. Second, the perpetrators engage in these activities while working for the state rather than acting as individuals. Third, state actors are corrupt because they do not protect and respect human rights. Therefore, corruption results from abusing power while acting on behalf of the state or violating the duty to respect, protect, and fulfill citizens' rights.

Three principles could be advanced and used as a theoretical basis for corruption as a direct violation of human rights. This means that certain rights are infringed as a result of corrupt activities (Adzanela 2011). For instance, the right to a fair trial is breached when a defendant bribes judges to not function impartially and independently. The rights to health and education are violated when people bribe doctors or principals of public schools or colleges to obtain health services or secure admissions (International Council on Human Rights 2009). Gebeye stated that corruption contributes to events that lead to the violation of human rights, meaning it is a sine qua non of the violation. Corruption is the primary cause of incidents and actions that constitute human rights abuses (Gebeye 2012). For instance, immigration officials may permit the importation and placement of dangerous goods close to populated regions. The materials violate the rights to health and life indirectly when residents become ill or die. Although corruption is not the root of the infringement on this right, it significantly contributes to the violations.

Corruption could be a remote violation or one of the contributing factors (Hermann and Warhurt 2009). For instance, demonstrations might occur due to an electoral offense that compromises the results' validity and accuracy. These protests could be ended using repressive tactics that violate human rights. The rights to live, not to be tortured, and freedom of expression might also be violated when state agents shoot, beat, and unjustly imprison demonstrators. However, corruption is only one of the key causes of human rights violations. Corruption violates the right to equality and non-discrimination. The International Covenant on Civil and Political Rights and the International Covenant on Economic, Social, and Cultural Rights indicate that the rights to life, liberty, the pursuit of happiness, and freedom from discrimination are fundamental. Each person has a right to equal protection and is entitled to equality before the law. This means that everyone should be treated without discrimination based on the right to equal treatment (Besson 2012). However, not all forms of prejudice and disparities in treatment result in discrimination. General international law principles stipulate that discrimination occurs when (a) the same case is handled differently, (b) a variation in behavior is not based on legitimate goals and reasons, or (c) there is an imbalance between the goal and the means employed in its achievement (Terracino 2007).

Everyone has the right to equal treatment from public officials while discharging their duties. In line with this, public officials accepting bribes enjoy special treatment compared to others. Human rights are violated because the same issue is handled differently due to

bribery that is not founded on legitimate objectives and causes. In particular, corruption significantly alters the equality of treatment and leads to discrimination. Its definition has four primary components that pertain to corruption. First, corrupt conduct results in a distinction, exclusion, or preference, while discrimination implies expenditures, limitations, or preferences. Second, discriminatory actions are taken based on a person's background, race, color, or gender. Third, the definition of discrimination prohibits behaviors with discriminatory intentions. In this regard, the nature of corruption includes discriminating objectives and results. Fourth, discrimination has negative effects that make it impossible to recognize, enjoy, or exercise other rights equally, including the right to life, education, and health. Several instances of corruption also give people preferences when exercising a right. For instance, students infringe on the right to non-discriminatory treatment when they offer the school money in exchange for admission.

Corruption infringes on the right to life, which every person is entitled to, and it needs a broad socioeconomic understanding. This right is violated when a public official is bribed to facilitate the unlawful importation of hazardous chemicals that kill people. Furthermore, corruption violates the freedom of religion when used inclusively and liberally to connote theistic, non-theistic, and atheistic beliefs and not to profess any of these ideas. The freedom of religion or belief needs to be recognized, protected, and fulfilled by the state (Tahzib 1996). Essentially, this right is violated when the government's budgetary allotment for establishing places of worship is tainted. The followers of the respective religion cannot establish places of worship and lack places for conducting their religious rites.

Corruption also violates the universal rights to freedom and security that should be respected and protected. The right to individual liberty is not unalienable and is not breached when the police are detained or arrested on legitimate legal grounds. Furthermore, the right to individual freedom affirms that everyone has a constitutionally guaranteed right to be free from arbitrary restrictions. All limitations on freedom should have legal justification, though this right could be violated unintentionally by corruption. For instance, a person aware of a corruption case could be dubbed a whistle-blower and be imprisoned, threatened with harm, or killed. Also, the right to personal security is violated when people face death threats for investigating and exposing corruption.

Corruption infringes on the freedom of movement—a crucial requirement for growth. Everyone has the right to freedom of movement, including living in any country and moving from one territory to another (Yalincak 2013). This correlates with the right to obtain travel documentation, in which discrimination occurs when the grounds for obtaining registration, an identity card, or a passport is subjective. Individuals with the same circumstances might be handled differently, leaving those without bribes in unfavorable conditions. In this case, the official is bribed, violating the person's right to freedom of movement. Corruption also violates the right to public information from governmental agencies, which everyone is entitled to. Government entities should respond and provide access to information based on this right unless there is a legal prohibition (Elsaman 2020). The right to public information is vital for the anti-corruption preventive system to be successful and coordinated. This is especially necessary for sectors directly tied to the public interest and vulnerable to abuse (Marzen 2020). The right is significant because it pertains to public oversight of governmental operations. Furthermore, public control through the right to engage in these domains is vital. Government money designated for education and health is susceptible to corruption by state officials; thus, it is necessary to ensure that the funds are used appropriately.

The right to access information should also be used when the public and the media's right to participate in the public arena are exercised. The corrupt actions of government officials cannot be stopped when the government does not grant the public access to information. The right to public information is expressed through transparency as a guiding principle of corruption-prevention policies and practices. The government may not follow this principle when people are not given the information required. Contrastingly, the principle of accountability states that the government must present accurate information

to the public. The right to equal access to public services is connected to accountability. Providing everyone, including the media, with equitable access to public services is vital to a successful anti-corruption preventive system. This right has been violated through bribery in managing the acquisition of public goods and services.

Corruption infringes the right to a fair trial, which entails procedural protections in the legal system or due process to ensure that justice is administered fairly, effectively, and efficiently (Bostan 2004). Everyone is entitled to equal treatment in court, as well as a just and public trial conducted by a competent, independent, and impartial judiciary (Sherman 2017). The right to a fair trial is especially important in the relationship between corruption and human rights and is violated when a judge is bribed to rule against one party. It is also directly violated when the public prosecutor is bribed to charge or acquit the defendant. Similarly, a defendant may use criminal threats to bribe investigators into using items, violating their right to a fair trial.

Corruption violates the right to the availability, accessibility, acceptability, and quality of public health and healthcare facilities, goods, services, and programs. Health facilities and services should be sufficient, accessible to everyone without discrimination, and of high quality, based on scientific or medical standards. Moreover, this right includes the elements determining its realization, such as access to safe and clean drinking water, proper sanitation, and a healthy environment (Terracino 2008). The state has to maintain and improve a clean and healthy environment for present and future generations (Kansman 2020). Regarding corruption, the state infringes on the right to health when it allots funds for the underprivileged to receive free treatment in a few designated hospitals. However, the number of deserved hospital directors is arbitrarily reduced, meaning the hospital's activities injure the state and violate the right of the poor to free medical care (Sekalala et al. 2020).

Corruption infringes on the right to education that everyone is entitled to as it supports the full development of human personality and dignity. Education should also be directed at strengthening respect, protection, and fulfilling human rights and fundamental freedoms (Lawler 2018). The four key components of education are availability, accessibility, acceptance, and adaptability (International Council on Human Rights 2009). The state is expected to provide free primary education to all residents. However, progressive and free secondary and higher education must also be made available and open to all individuals without hindrance or discrimination (Margerin 2010). Furthermore, the program, forms, and content of primary education should be free and accessible to students and parents. This should entail its quality, cultural appropriateness, relevance, and flexibility in response to societal demands (International Council on Human Rights 2009).

One element of the right to education is violated when a principal misuses the education funding meant for primary school children. This implies a corruption offense, which undermines the nation's finances or economy. As a result, students are forced to pay part of the fees to compensate for the corrupted payments. Infringing the right to education relates to the state's responsibility to make free and basic education available to all students. Furthermore, the principal may lower the price of secondary and higher education materials to support the planned program. As a result, the education program is incapacitated to meet the requirements set forth by the government. These acts constitute a crime that endangers state finances or the national economy, infringing on the right to an adequate education.

Corruption also violates the right to food or freedom from hunger (Kong 2009). Everyone's ability to feed themselves is guaranteed by their right to adequate nourishment (Burgess 2010). In this regard, food should be sufficiently accessible to meet each person's needs. However, it is important to ensure that harmful substances are not accessible to people according to their culture (McDermott 2012). An example of how food availability undermines someone's rights is bribing public officials to secure permits for property ownership and use (Nichols 2012). Also, the right to food is also violated when someone is

given more or better-quality agricultural land through bribery. This right is also infringed when land use is licensed based on bribes, resulting in uneven food distribution.

Corruption also violates the right to water, to which humans require access in order to survive and maintain a reasonable quality of life. Water is a precious and in-demand resource not accessible to everyone (Černič 2011). Therefore, the right to clean water could only be implemented by the common efforts of states, individuals, and other private and public institutions (Jankovic 2021). These parties must ensure the availability of sufficient water for personal and domestic uses. Water must also be of acceptable color, odor, and taste, as well as safe and sufficiently physically accessible (Satterthwaite 2021).

Water safety and hygiene help prevent dehydration-related deaths and lower the danger of contracting diseases. In general, the right to water encompasses water use for drinking, cooking, and health. Everyone needs equal access to sufficient, safe, acceptable, physically accessible, and affordable water for personal and domestic uses (Nemeth 2022). Therefore, the state should protect and improve the cleanliness of the environment for present and future generations (James R. May 2021). The scarcity of water resources is due to dishonest behavior and not the lack of access to clean water and increased air pollution (Ndeunyema 2020). In line with this, corruption violates the right to water when it restricts access to it. For instance, the right of local communities to own water was violated when firms bribed state water regulators to acquire excessive abstractions over rivers and waterbeds.

## 4. Results and Discussion

### 4.1. Corruption and Human Rights Violation in the United Nations Convention against Corruption, 2003

Indonesia ratified the United Nations Convention against Corruption 2003 (UNCAC) with Law Number 7 of 2006. The UNCAC is concerned about the seriousness of the problems and threats posed by corruption to social stability and security, undermining the institutions and democracy, ethical values, and justice, and jeopardizing sustainable development and the rule of law. The three essential terms related to human rights values include the rule of law, democracy, and sustainable development. The rule of law encompasses several elements, including the accountability of governmental actions, the proper use of discretionary authorities, and the protection of human rights (Kumar 2003). Although there are differences between human rights and democracy, respecting and upholding them is a fundamental principle of democracy (Lister 2012). Sustainable development has many different interpretations. However, it essentially refers to meeting present demands without compromising future generations' ability to do the same. This approach has consequences for transitioning from ecological sustainability to social and economic growth (World Commission on Environment and Development 1987). In this context, reducing corruption helps achieve sustainable development. While corruption is a problem, environmental corruption presents particular difficulties for sustainability (Bratspies 2018).

The definitions of law enforcement in the preamble to the UNCAC include an independent and impartial court, widely accepted laws, the application of the law to all people and organizations, and the use of force against an individual or group (United Nations Development Programme 2004). The UNCAC adopts a human rights-based perspective to create material content for preventing and countering corruption. The perspective begins by acknowledging the existence of civil, political, and economic rights as a human rights-based strategy to be promoted and safeguarded (Roht-Arriaza 2021). Article section (1) states that this Convention aims to promote the integrity, accountability, and good management of public affairs and property. Equitable access to public services and information contributes to preventing and countering corruption, which hinders good government and transparency (Al-Jurf 1990). Many governments fail to adhere to integrity and accountability in public affairs and wealth. They restrict access to information needed by interested parties, making it difficult to detect and combat corruption. Access to infor-

mation is crucial for preventing government activities that could lead to corruption. Goods, services, and government offices could be sold for profit because corruption includes the abuse of public wealth for personal benefit (Kumar 2004). This makes it challenging to uphold the government's principles of transparency and accountability.

Equal access to public services and information is crucial for preventing and combating corruption. For this reason, the UNCAC was created to improve the integrity, accountability, and good management of public affairs and wealth (Kumar 2004). Everyone has the right to seek, receive, save, and transfer publicly available information. Officials and parties involved in public procurement usually cover, protect, and create closed networks to pursue personal gains. Therefore, determining an unofficial relationship based on bribery could be challenging for both parties. This highlights the significance and necessity of the right to equal access to public information in procurement and financial management (Truelove 2003; Kiai 2007). For this reason, the UNCAC adopted a human rights-based approach as its key principle.

The UNCAC requested that participating states need to criminalize certain activities, beginning with bribery. Article 15 defines bribery as a promise, offer, or undue distribution of benefits to public officials directly or to other entities to act or not act in discharging their duties. According to Article 16, bribes may be given to national and foreign public officials as well as public international organization employees. International bribery occurring beyond national borders falls under the second category (Spahn 2009). In line with this, the UNCAC prohibits active and passive forms of bribery.

The second activity that needs criminalization is trading in influence. Section (1) of Article 18 defines trading in influence as promising, offering, or providing an undue benefit to public officials or other people directly or indirectly so they abuse their existing or perceived influence. The intention is to obtain an undue benefit from a government agency or public institution of the State Party. Furthermore, section (2) of Article 18 states that trading in influence occurs when public officials or other people make impropriety demands or receive benefits for themselves or for other people to abuse their actual or perceived influence. The goal is to obtain impropriety benefits from a government agency or public institution of the State Party. Trading in influence could be an active act when people promise, offer, or provide improper favors to public officials because of their power. A public official engaging in passive trading in influence accepts an improper promise, offer, or benefit from another person.

Trading in influence could violate the rights to political participation, equality, freedom of speech, expression, and information. Public officials may receive prominent coverage from a media outlet to promote the growth of a television network's broadcasting rights. This violates the freedoms of opinion, speech, and information, as well as the rights to equality and discrimination. Similarly, eliminating a portion of the permanent voter list supporting a president's spouse for one of the spouses to win the election violates the right to political participation. This occurs when public officials in the election commission act under the influence gained by accepting bribes from one of the candidates' spouses. Freedom of speech and expression is not limitless and requires precautions to ensure that it does not impinge on the rights and freedoms of others. Therefore, defamation laws are legal because they safeguard other people's reputations and rights (Trechsel 2000). Public authorities could pay journalists to publish false information or malicious statements concerning fraud committed by officials or others. This is an abuse of function and a violation of the freedoms of expression and information dissemination (Rajagopal 1999).

The third activity is illicit enrichment, which occurs when public officials increase their wealth substantially without explaining the rise in legitimate income (Article 20 of the UNCAC). Regarding corruption, public officials could accumulate wealth incommensurate with their income in many ways. This could directly harm the state's finances if they accept bribes from third parties in exchange for serving their interests, or if they embezzle money, securities, and other public property. The large increase in a public official's wealth is part of engaging in actions to unlawfully enrich themselves. The income could be

calculated based on the length of service and the presence or absence of business conducted while in the office. Public officials violate the right to property ownership by illegally enriching themselves with an unjustifiably enlarged wealth relative to their legitimate income. Everybody has the right to own property acquired legally, and the right is violated when the property is acquired illegally.

The fourth act that should be criminalized is bribery in the private sector. According to Article 21, it involves conducting economic, financial, or trade activities by promising, offering, or giving undue benefits to people working in private sector offices so they violate their duties. Also, bribery includes soliciting or accepting an unfair benefit from people in authority or working for a private sector organization to act or refrain from carrying out obligations.

Trading influence and bribery in the private sector are the new acts criminalized in the UNCAC. The topic of bribery is relevant to studies related to human rights concepts. According to Cecily Rose, corrupt practices in the private sector are not covered by international human rights documents such as the International Covenant on Economic, Social, and Cultural Rights. The only party legally liable for abuses of human rights is the state. The Covenant obligates states to defend human rights within their borders by passing laws that forbid private parties from violating the rights of others (Rose 2011). However, it does not specify the duties nations need to fulfill regarding avoiding corruption in the private sector. Current theories consider corrupt activities that affect the public and commercial sectors. The conservative view that the private sector should not be included in anti-corruption studies should be abandoned. This is because corrupt practices in the private sector are a part of the corruption spectrum. Many international organizations focus on preventing corruption in the private sector and designing appropriate policies.

Studies on corruption in the private sector intersect with upholding human rights by connecting two significant concerns. First, the state should safeguard its citizens from external actors and internal state agents that may violate their human rights. It needs to take proactive steps to ensure that people or organizations do not infringe on the human rights of its citizens. In this case, the state violates its human obligations when it does not take sufficient action or comply with the law to prevent, investigate, punish, or rectify losses brought on by the conduct (Terracino 2008). Second, the private sector is affected by privatization, which significantly impacts corruption and human rights. It facilitates transferring public services such as health, transportation, or telecommunications to the private sector by moving budget allocations and regulatory authorities, contributing to corruption (Terracino 2008).

In legal enforcement, combating corruption should emphasize prevention, elimination, and asset recovery. The two essential tasks associated with asset recovery are determining the property to be accounted for foreclosure and seizure. In this context, Article 31 of the UNCAC has emphasized several crucial elements, including (a) a breach of the convention that leads to the proof of property, (b) evidence of its application when the convention is broken, (c) wealth indicates the transformation, (d) evidence of the convention's infringement and the legal mixing of properties, and (e) evidence of the acquisition of wealth or profit.

Participating states need to prepare plans for confiscating property resulting from convention infractions. As a signatory to the UNCAC, Indonesia must practice the following fundamental tenets: (1) the government needs to create effective anti-corruption laws, (2) community involvement is necessary, and (3) international collaboration is crucial. Moreover, there is a need for activities that compensate states for financial losses sustained due to these crimes. Efforts to prevent and prosecute corrupt individuals are also necessary to eradicate corruption. The severity of the sentence for corruption perpetrators may be diminished when stolen assets are not returned. In line with this, it is difficult to retrieve stolen state property through corruption offenses. This is because the perpetrators have easy access, are hard to catch, and can conceal or launder the proceeds of their offenses. Additionally, recovery operations are hampered because the haven for the stolen money

extends outside the borders of the nation where the crime is perpetrated (Webb 2005; Vlasic and Cooper 2011).

The right to property is also associated with asset recovery. The state's illegal seizure of a person's property infringes the right to property ownership. Therefore, it is necessary to respect individual rights and minimize the profits received by the offenders from the corruption proceeds in the case of asset seizure. The UNCAC has made an effort to strike a compromise between these two interests (Kututwa 2007). Based on Article 31, assets obtained through corruption may be seized following a legally binding court ruling. This implies the necessity to establish beforehand that the assets being seized were acquired through corruption, used as tools to commit crimes, or purposefully combined with assets not acquired through the proceeds of corruption. The state does not violate the right to property ownership when all those conditions are met. In this case, everyone has the right to acquire property, but not through illegal means. This means that the state does not violate the right to property ownership by confiscating people's possessions acquired through corruption.

### 4.2. Law on Corruption Eradication, 1999/2001

Law 20 of 2001, which revised Law 31 of 1999 concerning corruption eradication, states that corrupt activities have caused human rights infringement. The widespread corruption has damaged the state's finances and violated the social and economic rights of the general public. Therefore, it should be defined as a crime that needs to be eradicated using a human rights-based approach. In this context, two factors make a human rights-based approach to crime prevention and eradication of corruption necessary. First, human rights law in international conventions is designed to shield all citizens from egregious social, legal, and political abuse. This idea forms the foundation for growth and has gained virtually universal acceptance. In this framework, one major topic in human rights studies is corruption. Second, corruption weakens fundamental rights, such as equality before the law, and non-discrimination, significantly contributing to their abuses (Koechlin 2007).

The Anti-Corruption Law lists seven categories of crimes. The first category is corruption associated with the loss of state finance under Section 1 of Article 2 and Article 3. This type of corruption could violate the rights to freedom of religion, life, food, health, education, workplace, and clean water. The rights to life and food are violated when a person corrupts a national or local government budget meant to provide for the food needs of the poor. Consequently, some poor people suffer from starvation, owing to a lack of access to food. The fulfillment of human rights to life, non-discrimination, and access to public information depends on the right to health (Ugaz Sanchez-Moreno 2007). In some instances, corruption in the health sector could lead to violations of the rights to life and health. The right to access information frequently overlaps with the right to health. In this case, prohibiting discriminatory treatment for access to healthcare demonstrates the connection between the right to health and non-discrimination (Toebes 2007).

The right to health includes access to appropriate healthcare and other rights that affect how well that right is met. These include access to clean water, personal hygiene, good sanitary conditions, and a safe environment (Bellinger and Sullivan 2022). Corruption violates the right to health by harming the state's budget meant for the poor to receive free treatment at certain designated hospitals. However, the hospital director unilaterally limits the number of the poor eligible for free care. Some patients also die when hospital personnel corrupt the machinery designed to clean the facility's water. In this case, the official's corruption practices violate the patient's right to life and health. Corruption also violates the fundamental right to education without sex discrimination, as is firmly established by international law. There may not be exceptions to this privilege during times of war, hostilities, or other emergencies. Education must be available to all citizens without hindrance or discrimination (Strayer 2019). It must also be flexible enough to change in response to societal demands (International Council on Human Rights 2009).

In a certain corruption case, the funding for education designated for pupils in primary schools was slashed by the principal. As a result, the students were required to pay a fee to compensate for the corrupted payments. This infringement of the right to education relates to the state's responsibility to make free, basic education available to all students. Another example is when principals embezzle funds to buy secondary and upper-education resources to support predefined curricula. This incapacitates the education program to meet the requirements set forth by the government, meaning the principals have infringed on the right to education accessibility. The right to work is also indirectly violated because the students face difficulties finding a job when they lack the educational requirements for the position.

Corruption also violates the right to access clean and safe water for drinking, cooking, and other domestic requirements, as well as preventing disease contamination and death from dehydration. Therefore, everyone should have equal access to physical and financial water services (Qureshi 2018). Decreased access to clean water and increased air pollution are caused by corruption, not a lack of water supply or management (Okaru-Bisant 2011; Terracino 2008). The central government may allot funds to purchase specialized equipment to clean the contaminated water. However, local regents may use the funds to purchase personal requirements, breaching the right to access water.

The second act is bribery, which is prohibited by the Anti-Corruption Law in Articles 5, 6, 11, 12, A, B, C, and D, as well as Article 13. Bribery refers to giving state officials or other public servants gifts or promises in exchange for their services. The offering of gifts or promises must be connected to one's position, linking bribery to the unlawful use of official authority in public offices. Furthermore, bribery may be active or passive and is mostly received by judges, attorneys, prosecutors, and police, as well as non-law enforcement, such as state apparatuses and public workers. Law enforcement, government officials, civil workers, as well as private parties, such as individuals or corporations, are all capable of accepting bribes. The judges and attorneys accepting bribes are punished based on paragraph 2 of Article 6 and Letters A and B of Article 12.

Giving or receiving gifts and promises violates civil–political, economic, social, cultural, and solidarity rights. Bribery violates the civil–political rights to a fair trial, equality and non-discrimination, life, protection from child labor, freedom of expression and from slavery or servitude, as well as the rights of minority groups. The economic and social rights violated due to bribery include the rights to just and favorable working circumstances, adequate housing, health, and education. Furthermore, bribery violates the solidarity rights to water and a healthy environment. Defendants violate the rights to a fair trial, equality, and non-discrimination when they pay judges to render decisions that exonerate or free them from the prosecution's requests (Jennett 2007). These two rights are also violated by defendants paying off the prosecutor to present the evidence that proved them (Gruenberg and Biscay 2007). Additionally, the rights to a fair trial and equality are violated by advocates soliciting presents or promises from opposing parties to support them.

State officials or civil servants violate the constitutional rights to health and adequate housing by taking bribes to approve the disposal of poisonous and public-health-detrimental waste in an inhabited area. Indirect violations of the right to life occur when residents die from this poisonous waste. Similarly, the rights to protection from human trafficking, sexual exploitation, forced child labor, and freedom from slavery or servitude are violated when immigration or police officials take bribes to help in transporting underage women abroad to work as sex workers or engage, undetected, in sexual exploitation. The rights of minority populations may be infringed when a petroleum business bribes state officials or civil servants to build oil pipelines through sacred locations belonging to indigenous peoples. Employers violate the rights of minority groups when they seize land by bribing government officials. The rights to freedom of movement, equality, and non-discrimination are also infringed when an official state document, such as a passport or visa, is obtained by paying an immigration officer a specific sum of money. This is due to limited access to official government papers or restrictions on international travel.

The right to a fair and enjoyable workplace is infringed when an employer bribes a labor inspector to avoid enforcing legislation. Similarly, the right to an education is violated when parents are forced to pay the principal money before their children can enroll in a primary school. A corporation providing clean water may propose an unapproved payment by the water regulator to overestimate the maximum volume of water to be delivered, violating the right to water. Furthermore, the right to environmental health and cleanliness is violated when a firm pays off the regent to grant a forest conversion permission against the law. In this case, the woodland to which the regent delegates its function is a protected forest and the center of the neighborhood. The corporation converts a previously protected forest into an oil-palm plantation area after the regent's authorization. Therefore, the regent's acceptance of bribes to provide forest conversion permits infringes on the nearby people's right to a healthy and clean environment.

The third act is forgery and embezzlement, implying doing anything to an object without its will or violating its ownership rights (Rusydianta 2020). Several circumstances involving the theft of cash, securities, products, deeds, or registers may constitute human rights breaches. For instance, the right to education may be violated when authorities embezzle money designated for education programs. This diminishes the availability of physical and non-physical facilities, reducing the quality of schooling (Terracino 2008). Children without access to education would have difficulties developing their personalities, talents, and mental and physical capacities. Education or in-person instruction helps people to know their rights, including accessing information, practicing their religion or philosophy, being free from discrimination, and having good health (Huck 2012). In another scenario, the right to health is violated when a county hospital treasurer or director assumes ownership of a car assigned to the hospital for dropping off and picking up underprivileged patients. In this case, the right to health for the underprivileged is infringed when the director embezzles funds from 4 of the 20 cars at the local hospital.

Forgery refers to altering the writing of published books or lists to make the information false (Gani and Rahaditya 2020). This could involve adding or deleting one or two words or numbers or creating something new with a purpose and meaning different from the original. For instance, a regional hospital's director violates the right to health by falsifying the number of low-income people that have used their right to free treatment. The list may indicate that ten thousand patients received free therapy from September to December. However, the director may write twenty thousand instead, giving fewer impoverished people free medical care. This misappropriation may limit poor people's access to free medical treatment, forcing some low-income patients to cover the costs of their visits.

The fourth act is the unlawful extortion prohibited by Articles 12 E, F, and G. State officials or public servants abusing their positions might compel someone to work for them, pay something, or take money with deductions. Corruption through counterfeiting violates the right to equality before the law. For instance, a driver that disregards traffic laws may be detained by a police officer. Despite the officer's best efforts, the driver may want to be legally ticketed and have no interest in making amends. When the person does not agree to pay the required money, the police may threaten to tow their automobile, violating the right to equality before the law. In another case, the needed charge was only USD 300, but immigration officers informed other agencies' civil servants that they had to pay USD 600. An applicant for a passport also declined to pay the requested USD 600 instead of USD 300. The immigration officer stated that the additional USD 300 was owed by one of the applicant's coworkers that neglected to pay a processing charge of USD 300. In this case, the right to equality before the law was violated because the other passport applicant owed the immigration official nothing.

The fifth act is cheating, which is prohibited in Article 7. A person's fraudulent activities could infringe on the rights to life and personal security. For instance, the right to life was infringed when the government contracted someone to construct a 2 km bridge. According to the agreement, one million bags of the highest-grade cement were needed to

build the bridge. Three months after the construction, the bridge collapsed. A thorough study by experts showed that 800,000 bags of third-class cement were used, contributing to the collapse. When the bridge collapsed, resulting in 20 fatalities and 200 injuries, the contractor's dishonesty directly violated the rights to life and personal security.

The sixth act is a conflict of interest during the acquisition of items, as prohibited in Article 12 i. This relates to civil officials knowingly participating in managing or overseeing the contracting, procurement, or renting of the deed. The right to equality and non-discrimination is violated when the person is involved directly in purchasing goods and services. In this case, the procurement committee gives participants and overseers preferential treatment for goods and services. The civil servant must ensure that the bridge procurement procedure conforms with applicable rules and regulations. This is because the civil servant manages the procurement process for the connecting bridge between districts A and B. A conflict of interest exists when a corporation is also one of the players in the procurement process. It is challenging to establish equitable rights among all participants because the civil servant's company would typically win the procurement, violating the right to equality and non-discrimination (International Council on Human Rights 2009).

The seventh act is receiving gratuities, which is prohibited in Articles 12B and 12C of the Anti-Corruption Law. Gratification includes obtaining domestic or overseas cash, commodities, rebates or discounts, commissions, interest-free loans, airline tickets, accommodation, vacations, free medical care, and other amenities using technology or other means. The right to equal access to public services is violated when civil servants accept a gratification dealing with their position but contradict the commitments or duties assigned. For instance, an immigration officer may receive a laptop as a gift from a visitor at one of his daughters' weddings. The officer may know that the guest is one of the applicants for a passport denied because the standards were not met. In this case, the recipient must report the laptop gift to the Corruption Eradication Commission (KPK) immediately. Based on Article 12C of the Anti-Corruption Law, this is a bribe and a violation of the right to equal access to public services when the officer waits up to 30 days after receiving the laptop before reporting it to the KPK.

*4.3. Judicial Decisions*

This study analyzed 38 verdicts in corruption cases concerning human rights violations. Most verdicts dealt with the corruption of public finances and extortion perpetrated by judges, civil workers, state apparatuses, advocates, and private parties. The findings showed that only nine corruption case verdicts directly violated human rights. Table 1 shows the relationship between corruption, human rights violations, and the severity of punishment in court decisions.

**Table 1.** The Relation between Corruption and Human Rights Violations, as well as the Severity of Punishment in the Court Decisions.

| Court Decision | Case Position | Human Right Violation | Severity of Punishment |
| --- | --- | --- | --- |
| 11/Pid.B/TPK/2006/PN.Jkt/Pst | The defendant received USD 660,000 from witness Artalyta Suryani for providing updates on the investigation into the alleged corruption case, which was confidential and allowed Sjamsul Nursalim to be heard as the former President and Director of BDNI. Tbk not involved in the process | Rights to non-discrimination | 20 years of imprisonment and an IDR 500 million fine |
| 04/Pidsus/TPK/2011/Pn.Srg | The defendant accepted bribes from Ferry Priatman Hakim of IDR 500,000,000 to avoid being named a suspect | Right to a fair trial | 1.5 years of imprisonment and IDR 20 million fine |

**Table 1.** *Cont.*

| Court Decision | Case Position | Human Right Violation | Severity of Punishment |
|---|---|---|---|
| 14/Pid.B/TPK/2010/Pn.Jkt.PSt | The defendant had received IDR 300,000,000 from witness Adner Sirait in his capacity as counsel from witness Darianus Longguk Sitorus, President Director of Sabar Ganda Tbk | Right to a fair trial | 6 years of imprisonment and an IDR 200 million fine |
| 17/Pid.B/TPK/Pn.Jkt.Pst | The defendant makes false of the Education Office's budget to his advantage | Right to education | 7.5 years of imprisonment and IDR 300 million fine |
| 54/Pid.B/TPK/2012/P.Jkt.Pst | The defendant accepts bribes from Mindo Rosalina Manulang in connection with the project to secure educational facilities at the State University | Right to education | 12 years of imprisonment and an IDR 500 million fine |
| 13/Pid.B/2008/PN.Jkt.Pst | The defendant accepted bribes related to converting the forest area into a business area | Right to a healthy and clean environment | 2.5 years of imprisonment and IDR 100 million fine |
| 31/Pid.B/TPK/2010 /Pn.Jkt.Pst | The defendant, the Head of the Planning and Budget Bureau of the Secretariat General of the Ministry of Health of the Republic of Indonesia, corrupted state funds in the procurement project of portable X-ray equipment for servants and local hospitals | Right to a clean and healthy environment | 5 years of imprisonment and an IDR 100 million fine |
| 22/Pid.B/TPK/2008/PN.Jkt/Pst | The defendant took payments regarding the process of releasing protected forest lands | Right to a clean and healthy environment | 4.5 years of imprisonment and IDR 200 million fine |
| Nomor.123/Pid.Sus-TPK/2017/PN. Jkt. Pst), | The defendant issued a mining permit to AHB Tbk to conduct exploration activities and production operations to the detriment of state finances | Right to a clean and healthy environment | 12 years of imprisonment and an IDR 1 billion fine |

Sources: proceeded by the author.

Table 1 shows that the judge's legal reasoning directly connected the defendants' corruption with human rights violations. The defendants violated rights to a fair trial, non-discrimination, education, health, and a clean and healthy environment. However, the sentence imposed by the judges does not directly correlate with the crime committed. Offenders received sentences of more than ten years in jail only in three cases, while the rest were sentenced to less than eight years. Therefore, the aggravating factor in sentences is not always the direct violation of human rights because of the defendant's misconduct, as several judges only gave prison terms of 1.5 or 2.5 years.

The court did not impose harsh prison terms even when the defendant's corrupt acts directly violated human rights. This is because human rights violations are not included in the aggravated criteria, heightening the criminal sanction. Only two of the nine verdicts explicitly mentioned human rights violations as aggravating considerations. The first is the corruption case against former House of Representatives member Angelina Patricia Pinkan Sondakh. In this case, the defendant was adjudged to have accepted a bribe of IDR 12.5 billion from Permai Grup Tbk for building athletes' estates and the facilities of public universities. One aggravating circumstance is that the defendant's conduct infringed on the community's economic and social rights, including the right to education. Therefore, judges connected corruption to human rights violations even when using the general term (Court Decision Number 54/Pid.B/TPK/2012/P.Jkt.Pst (Pengadilan Negeri Jakarta Pusat 2012)) and sentenced the defendant to 12 years.

The second corruption case involved Urip Tri Gunawan, the former public prosecutor at the Directorate of Investigation for Special Crimes at the Attorney General's Office of the Republic of Indonesia. The defendant was paid USD 660,000 by the witness Artalyta Suryani to provide information about the progress of the investigation into alleged confidential corruption cases. Therefore, the defendant gave Sjamsul Nursalim, the former President and Director of BDNI Tbk, a chance to not present during the suspected corruption crime investigation. In one of the aggravating factors to punishment, the defendant had engaged in acts of discrimination in law enforcement during the investigation of Sjamsul Nursalim's BLBI II of BDNI Tbk case (Court Decision:11/Pid.B/TPK/2006/PN.Jkt/Pst (Pengadilan Negeri Jakarta Pusat 2006)).

In this context, the theories of punishment, such as proportionality theory, prevention theory, and rehabilitation theory, which result in criminal individualization, cannot be separated from the criminal punishment that courts administer. Retribution theory is now used in the context of penal proportionality. According to this view, there must be a direct correlation between the seriousness of a crime and the severity of the punishment (Luna 2003). The threat of criminal punishment to the perpetrator increases with the severity of the offense (Goh 2013). Criminal propriety relates to the proportionality concept, meaning that crime seriousness functions as the parameter to establish the severity of the criminal punishment (Husak 2011). Mildly significant crimes should not be penalized with more offenses (Husak 2020). According to Hudson, the simple way to establish proportional punishments is by 'ranking offences according to seriousness and then establishing a scale of penalties of commensurate severity' (Hudson 1996).

Prevention theory argues that criminal sanctions are imposed to prevent both the offender and potential offenders from repeating or committing the crime in the future (Sarma 2017). The costs suffered by the offender after performing a criminal act must be larger than the advantages acquired in order for criminal punishments to successfully deter criminals from committing crimes (Cicchini 2010). Criminal penalties must be harsher than the seriousness of the offense (Barnes 1999). Prevention theory takes a different perspective, in contrast to rehabilitation theory, which emphasizes the individualization of penal sanction. Criminal penalties should be tailored to the circumstances of the offender and the specifics of the crime committed. As a result, the only way that rehabilitation programs may lower crime rates is through flexible punitive laws. In this sense, abusers who are drug addicts are considered, and other treatment programs must be tailored to the needs and characteristics of the offender. This program unmistakably calls for the individualization of penal sanctions (Tonry 2006).

Criminal aggravation for corrupt individuals should take into account not only the rights of the victims but also the rights of the perpetrator of corruption, the peculiarities of corrupt criminal acts, and the goal aimed at this aggravation. The concept of the individualization of penal sanctions and prevention must be considered as a criterion to increase the criminal threat to corrupt officials, making jail the very last resort. Society as a whole is a victim of corruption, and what it requires is the recovery of state financial losses that have been tainted by offenders, in order to improve healthcare prices, educational facilities, and the economic income. As a result, the theory of punishment for aggravating the severity of penal sanctions for offenders no longer relies on the theory of proportionality; rather, it adopts the theories of prevention and rehabilitation. This last theory is thought to take into account both the human rights of corrupt officials as well as the human rights of crime victims. Under a human rights perspective, the state has an obligation to pay attention to all human rights, including the rights of convicted persons of corruption cases. This is stipulated in Article 2 of the Universal Declaration of Human Rights from 1948 (UN): "Everyone is entitled to all the rights and freedoms set forth in this Declaration, without distinction of any kind, such as race, color, sex, language, religion, political or other opinion, national or social origin, property, birth or other status. Furthermore, no distinction shall be made on the basis of the political, jurisdictional or international status of the country or

territory to which a person belongs, whether it be independent, trust, non-self-governing or under any other limitation of sovereignty".

## 5. Conclusions

The United Nations Convention against Corruption 2003 and the Anti-Corruption Act 1999/2001 have incorporated a human rights perspective into handling corruption cases. The law refers to corruption as an infringement on people's economic, social, and cultural rights. This study found a link between corruption in Indonesian law and human rights abuses. It discussed the development of such partnerships using a realistic scenario and comprehensively described the nexus between corruption and human rights violation. However, the results indicated that there is insignificant evidence of this connection in corrupt court judgments. In their legal analysis, few court decisions link corruption to human rights violations. Judges also consider the relationship more thoroughly when making legal considerations and it is not applied as an aggravated circumstance, resulting in significantly milder prison sentences.

This study makes several recommendations. First, corruption has weakened the foundation of democracy and violated economic, social, cultural, civil, and political rights. However, the public's response to this issue has not been positive. Therefore, academics, practitioners, policymakers, and civil society organizations should consider corruption a violation of human rights. Second, the supreme court, the attorney general, the police, and the Corruption Eradication Commission should jointly issue guidelines for applying corruption as a violation of human rights. The obligation for investigators, public prosecutors, and judges to identify and establish links between corruption cases and human rights violations should also be regulated in this legal provision. In this case, human rights violation is an aggravating factor of harsh punishment. Offenders infringing on more human rights should receive a more severe punishment, moving from the theory of penal proportionality to the theories of prevention and rehabilitation. Additionally, judges or public prosecutors need to present human rights legal experts during corruption cases to clarify and find the nexus between corrupt acts and human rights violations. Third, corruption cases are currently prosecuted by the Attorney General's Office and the Corruption Eradication Commission but are tried by special corruption judges. The training provided also includes a stronger emphasis on comprehending corruption crimes. Therefore, it is essential to provide specialized training on prosecuting and proceeding with corruption cases by connecting them to human rights infringement.

**Author Contributions:** Conceptualization, M.A. and S.N.; methodology, A.M.; software, M.A.; validation, M.A., S.N. and A.M.; formal analysis, M.A.; investigation, S.N.; resources, M.A.; data curation, A.M.; writing—original draft preparation, M.A.; visualization, S.N.; supervision, A.M.; project administration, S.N.; funding acquisition, M.A. All authors have read and agreed to the published version of the manuscript.

**Funding:** This research received no external funding.

**Institutional Review Board Statement:** Not applicable.

**Informed Consent Statement:** Not applicable.

**Data Availability Statement:** Not applicable.

**Conflicts of Interest:** The authors declare no conflict of interest.

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
