# Peer review of "The Application of a Human Rights Approach toward Crimes of Corruption: Analyzing Anti-Corruption Regulations and Judicial Decisions"

_laws, 2014_

Round 1
Reviewer 1 Report
The article is good.
The article has a good literature review and empirical research, even though the cases were taken from a secondary source.
I suggest minor adjustments:
1) In the abstract, the author must also inform the methodology.
2) The scope is too broad (relationship between Indonesian anti-corruption laws and human rights violations). The author could delimit it a little more.
3) When referring to other texts, the article cites twice the same author. For example, lines 33-35 could read as follows: “Prabowo (2014) emphasized the importance of understanding how opportunities, pressure, and justifications for corruption are presented to potential offenders that often weigh the perceived rewards and drawbacks".
4) In line 108, I believe there was some mistake in the Latin expression “sine qou none”.
5) From item 3.4., the article brings a very interesting empirical research. It analyzed 38 verdicts in corruption cases concerning human rights violations. However, it was unclear how these 38 cases were selected, or if there are only 38 cases in total.
6) In conclusion, the author could link the empirical research and the theoretical approach more closely.
7) Reference Section (line 711) - The author must adapt with the Laws guidelines for references. See: https://www.mdpi.com/authors/references.
The quality of English is good. The article needs minor revision.
Author Response
Dear reviewer 1
Thank you for your review notes. Below are our responses to your insightful comments and suggestion.
1) In the abstract, the author must also inform the methodology.
We have added the methodology in the abstract (line 7-8 of the manuscript):
This study aimed to examine the connection between crime of corruption and human rights violations. Indonesian’s corruption eradication regulations have increased the possibility of handling human rights-based corruption cases. This study employed doctrinal legal research that mainly relied on the anti-corruption legislations and corruption cases in the judicial decisions. The results showed that the law states that corruption infringes on people's economic, social, and cultural rights. It employed a plausible scenario to provide practical explanations of the relationship between the two variables. The types of crimes of corruption have direct nexus to the violation of human rights. In addition, there was inadequate proof of the connection between corruption and human rights violations in court rulings. Specifically, a few court decisions relate corruption to human rights violations. Judges consider the relationship more thoroughly when making legal considerations and is not applied as an aggravated circumstance, resulting in significantly milder prison sentences. The findings imply the vitality of mainstreaming corruption as human rights violations through comprehensive and massive studies. Furthermore, legal enforcement institutions need to issue guidelines and provide continuous training on handling human rights-based corruption cases to the police, public prosecutors, and judges.
2) The scope is too broad (relationship between Indonesian anti-corruption laws and human rights violations). The author could delimit it a little more.
We have delimited the scope of this study as shown in line 47-56 of the manuscript:
The present study is aimed at analyzing the nexus between anti-corruption regulations as well as court decisions in Indonesian and human rights violations. It specifically examined how the types of corruption directed toward the government officials and other public servants in both United Nations Convention against Corruption 2003 (UNCAC) and Anti-Corruption Act of 1999/2001 infringe the human rights. Furthermore, the study examined the judicial responses to the need to apply the human rights approach. The findings of this study contribute to developing an effective corruption strategy on how to incorporate human rights approach in the substantive criminal law and criminal procedure. This can also be as the guidance for the judges in aggravating the severity of punishment in corruption cases.
3) When referring to other texts, the article cites twice the same author. For example, lines 33-35 could read as follows: “Prabowo (2014) emphasized the importance of understanding how opportunities, pressure, and justifications for corruption are presented to potential offenders that often weigh the perceived rewards and drawbacks".
We have modified the way to cite article as shown in line 38-46 of our manuscript:
Prabowo (2014) emphasized the importance of comprehending how opportunities, pressure, and justifications for corruption are presented to potential offenders that often weigh the perceived rewards and drawbacks. Moreover, Hadiprayitno (2010) analyzed human rights enforcement in the legal and political system. Juwita (2017) examined how international human rights legislation perceived the relationship between corruption and the right to education. Subsequently, Ersan established that legal understanding and relevant concepts that link human rights and the criminal process helps combat corruption (Ersan 2018). According to Farida (2022), confiscating an asset obtained through corruption does not violate human rights.
5) From item 3.4., the article brings a very interesting empirical research. It analyzed 38 verdicts in corruption cases concerning human rights violations. However, it was unclear how these 38 cases were selected, or if there are only 38 cases in total.
We have added the explanation on how the 38 cases were selected as shown in line 78-86 of the manuscript:
This study also examined corruption judicial decisions. To prevent the length of explanation and/or analysis, this study collected and reviewed 100 verdicts in corruption cases obtained randomly through Directory of Supreme Decisions (putusan3.mahkamahagung.go.id). However, 38 verdicts dealt with the misappropriation of public finances and extortion perpetrated by judges, civil workers, state apparatuses, advocates, and private parties that infringe human rights violation directly or indirectly. Only nine verdicts showed that corruption directly violates human rights. This data could be used to propose a comprehensive strategy to fight corruption through judicial decisions.
6) In conclusion, the author could link the empirical research and the theoretical approach more closely.
We have added such suggestion in line 699-703
Thank you.
However, the results indicated insignificant evidence of this connection in corrupt court judgments. In their legal analysis, few court decisions link corruption to human rights violations. Judges also consider the relationship more thoroughly when making legal considerations and is not applied as an aggravated circumstance, resulting in significantly milder prison sentences.

Reviewer 2 Report
Hello,
The theme is highly actual, and I recommend the author(s) continue research on this topic.
Statement of the issue is explained good. But I can’t see novelty in the article. This research needs to a mechanism as a solution with interdisciplinary approach.
Solution may be researched even for one area like sport or education or justice or …
The article needs to be revised and then review again.
Quality of English language is acceptable, just some irritation on terminology and academic text is needed.
Author Response
Dear Reviewer 2.
Thank you for your review notes. We have adjusted the paper on the basis of your review.
The comment of reviewer 2: The theme is highly actual, and I recommend the author(s) continue research on this topic. Statement of the issue is explained good. But I can’t see novelty in the article. This research needs to a mechanism as a solution with interdisciplinary approach.
Our response has been added in line 37-56:
However, studies on the relationship between corruption and human rights abuse are limited to existing legal regulations. Prabowo (2014) emphasized the importance of comprehending how opportunities, pressure, and justifications for corruption are presented to potential offenders that often weigh the perceived rewards and drawbacks. Moreover, Hadiprayitno (2010) analyzed human rights enforcement in the legal and political system. Juwita (2017) examined how international human rights legislation perceived the relationship between corruption and the right to education. Subsequently, Ersan established that legal understanding and relevant concepts that link human rights and the criminal process helps combat corruption (Ersan 2018). According to Farida (2022), confiscating an asset obtained through corruption does not violate human rights.
The present study is aimed at analyzing the nexus between anti-corruption regulations as well as court decisions in Indonesian and human rights violations. It specifically examined how the types of corruption directed toward the government officials and other public servants in both United Nations Convention against Corruption 2003 (UNCAC) and Anti-Corruption Act of 1999/2001 infringe the human rights. Furthermore, the study examined the judicial responses to the need to apply the human rights approach. The findings of this study contribute to developing an effective corruption strategy on how to incorporate human rights approach in the substantive criminal law and criminal procedure. This can also be as the guidance for the judges in aggravating the severity of punishment in corruption cases.
Warm regards

Reviewer 3 Report
The author should consider providing a section on literature review. Besides the methodology seems to lack a lot of inputs and explanation.
None
Author Response
Dear reviewer 3.
Thank you for your great comments and suggestion for the improvement of the quality of our paper.
The Comments of reviewer 3: The author should consider providing a section on literature review. Besides the methodology seems to lack a lot of inputs and explanation.
We have made the literature review in line 87-272 of our manuscript. Also, we have added the methodology in more detail information in line 69-86:
This study employed doctrinal legal research relying on anti-corruption legislations and judicial decision on corruption cases. Primary data were collected from the Indonesian regulatory framework on the types of corruption as promulgated in the United Nations Convention against Corruption 2003 and the Anti-Corruption Act of 1999/2001. At least four and seven types of corruption have been criminalized in the Convention and the Act, respectively. The analysis of these laws focused on the types of crimes of corruption and their nexus to the infringement of human rights followed by descriptive examples. Additionally, the court rulings were highlighted to identify the decisions’ authenticity to provide sufficient legal considerations to apply the human rights approach to aggravate the severity of punishment. This study also examined corruption judicial decisions. To prevent the length of explanation and/or analysis, this study collected and reviewed 100 verdicts in corruption cases obtained randomly through Directory of Supreme Decisions (putusan3.mahkamahagung.go.id). However, 38 verdicts dealt with the misappropriation of public finances and extortion perpetrated by judges, civil workers, state apparatuses, advocates, and private parties that infringe human rights violation directly or indirectly. Only nine verdicts showed that corruption directly violates human rights. This data could be used to propose a comprehensive strategy to fight corruption through judicial decisions.
Regards

Round 2
Reviewer 2 Report
In this stage, i recommend to publish the current article.
In this stage, i recommend to publish the current article.